



# 1 Estimating flood damage in Italy: empirical vs expert-based
# 2 modelling approach

Mattia Amadio[1], Anna Rita Scorzini[2], Francesca Carisi[3] , Arthur H. Essenfelder [1], Alessio
Domeneghetti[3], Jaroslav Mysiak[1], Attilio Castellarin[3]
1 CMCC Foundation - Euro-Mediterranean Center on Climate Change and Ca' Foscari University of Venice, Italy
2 Department of Civil,Environmental and Architectural Engineering, University of L'Aquila, L'Aquila, Italy
3 DICAM, Water Resources,, University of Bologna, Italy
Correspondence to: Mattia Amadio (mattia.amadio@cmcc.it)
**Abstract**
Flood risk management generally relies on economic assessments performed using flood loss models of
different complexity, ranging from simple univariable to more complex multivariable models. These latter
accounts for a large number of hazard, exposure and vulnerability factors, being potentially more robust
when extensive input information is available. In this paper we collected a comprehensive dataset related to
three recent major flood events in Northern Italy (Adda 2002, Bacchiglione 2010 and Secchia 2014), including
flood hazard features (depth, velocity and duration), buildings characteristics (size, type, quality, economic
value) as well as reported losses. The objective of this study is to compare the performances of expert-based
and empirical (both uni- and multivariable) damage models for estimating the potential economic costs of
flood events to residential buildings. The performance of four literature flood damage models of different
nature and complexity are compared with the performance of univariable, bivariable and multivariable
models empirically developed for Italy and tested at the micro scale based upon observed records. The uni-
and bivariable models are produced testing linear, logarithmic and square root regression while
multivariable models are based on two machine learning techniques, namely Random Forest and Artificial
Neural Networks. Results provide important insights about the choice of the damage modelling approach
for operational disaster risk management.
**Key-words:** flood risk assessment empirical expert-based model machine learning stage damage curves

## 26 1. Introduction

Among all natural hazards, floods historically cause the highest economic losses in Europe (EEA 2010;
EASAC 2018). In Italy alone, a country with the largest absolute uninsured losses among EU countries
(Alfieri et al., 2016; EEA, 2016; Paprotny et al., 2018), around EUR 4 billion of public money were spent over
a 10 years period to compensate the damage inflicted by major extreme hydrologic events (ANIA 2015).
From 2009 until 2012, the recovery funding amounted to about EUR 1 billion per year; a fraction of the total
estimated damage of around EUR 2,2 billion (Zampetti et al., 2012). In this context, and particularly
compelled by the EU Flood Directive (2007/60/EC), sound and evidence-based flood risk assessments should
provide the means to support the development and implementation of cost-effective flood risk reduction
strategies and plans.



Several different approaches of varying complexity exist to estimate potential losses from floods, depending
mainly on the category of damage (e.g. direct impacts or secondary effects, tangible or intangible costs, etc.)
and the scale of application (i.e. macro, meso or micro scale) (Apel et al., 2009; Carrera et al., 2015; Hallegatte,
2008; Koks et al., 2015; de Moel et al., 2015). Direct tangible damages to assets are typically assessed using
simple univariable models (UVMs) that rely on deterministic relations between a single descriptive variable
(typically maximum water depth) and the economic loss mediated by the type/value of buildings or land
cover directly affected by a hazardous event (Huizinga et al., 2017; Jongman et al., 2012; Jonkman et al., 2008;
Merz et al., 2010; Messner et al., 2007; Meyer and Messner, 2005; de Moel and Aerts, 2011; Scawthorn et al.,
2006; Smith, 1994; Thieken et al., 2009). Empirical, event-specific damage models are developed from
observed flood loss data. A major drawback of empirically-based damage models relies on its low
transferability to other study areas or regions, as significant errors are often verified when these are used to
infer damage in other regions than those for which they were built to (Amadio et al., 2016; Apel et al., 2004;
Carisi et al., 2018; Hasanzadeh Nafari et al., 2017; Jongman et al., 2012; Merz et al., 2004; Scorzini and Frank,
2015; Scorzini and Leopardi, 2017; Wagenaar et al., 2016). Synthetic models, on the other hand, are based on
''what-if analyses'', relying on expert-based knowledge in order to generalise the relation between the
magnitude of a hazard event and the resulting economic damage. An advantage of synthetic models over
empirically-based models relies on the fact that the first are less sensitive to the input data, thus being better
suited for both temporal and spatial transferability (Smith 1994; Merz et al. 2010; Dottori et al. 2016).
Both empirical and synthetic models can be configured as uni- or multivariable. The vast majority of
univariable flood damage models account for water depth as the only explanatory variable to explain the
often complex relation between the magnitude of a flood event and the resulting damages; however, a non-
exhaustive literature search shows that other parameters may influence the flood damage process, such as
flow velocity (Kreibich et al., 2009), flood duration, and water contamination (Molinari et al., 2014; Thieken
et al., 2005), to name just a few. In addition, a large number of other non-hazard factors can be significantly
different from one place to another, such as type and quality of buildings, presence of basements, density of
dwellings, early warning systems and precautionary measures (Cammerer et al., 2013; Carisi et al., 2018;
Figueiredo et al., 2018; Kreibich et al., 2005; Merz et al., 2013; Penning-Rowsell et al., 2005; Pistrika and
Jonkman, 2010; Schröter et al., 2014; Smith, 1994; Thieken et al., 2008; Wagenaar et al., 2017b). Therefore,
multivariable models (MVMs) are potentially better-suited alternatives to describe the complex flood-
damage relation (Apel et al., 2009; Elmer et al., 2010). Common techniques applied in a context of MVM are
machine learning (e.g., Artificial Neural Networks and Random Forests) (Merz et al. 2013; Spekkers et al.
2014; Kreibich et al. 2017, Carisi et al. 2018), Bayesian networks (Vogel et al., 2013), and Tobit estimation (Van
Ootegem et al., 2015). Moreover, some MVMs support probabilistic analyis of damage (Dottori et al., 2016;
Essenfelder, 2017; Wagenaar et al., 2017a). MVMs need to be validated against empirical records in the



region of the model application in order to produce reliable estimates (Hasanzadeh Nafari et al., 2017; Molinari et al., 2014, 2017; Scorzini and Frank, 2015; Zhou et al., 2013). However, greater sophistication of MVMs requires more detailed hazard, exposure and losses description. Due to the lack of consistent and comparable observed flood data, this kind of models are still seldom applied. This is why it is necessary to compile comprehensive, multivariable datasets with detailed catalogue of flood events and their impacts (see Amadio et al., 2016, Molinari et al., 2012 and 2014, and Scorzini and Frank, 2015).

Our study contributes to this end by assembling detailed data on three recent flood events in Northern Italy. For each event, our dataset comprises the following building-scale data: 1) hazard characterization derived from observational data and/or hydraulic modelling, 2) high-resolution exposure in terms of location, size, typology, economic value, etc. obtained from multiple sources, and 3) declared costs per damage categories. Building upon this extensive dataset, we employ supervised learning algorithms to explore the parameters of hazard, exposure and vulnerability and their influence on damage magnitude. We test linear, logarithmic and square root regression to select the best fitting Uni-Variable (UVM) and Bi-Variable (BVM) models, and two machine learning techniques, namely Random Forest (RF) and Artificial Neural Networks (ANN) for training and testing the empirical MVMs. The models developed on the three considered case studies provide a benchmark to test the performance of four literature models of different nature and complexity, specifically developed for Italy. The results of this study provide important insights to understand the feasibility and reliability of flood damage models as practical tools for predictive flood risk assessments in Italy.

## 2. Study area

With an extent of 46,000 km², the Po Valley is the largest contiguous floodplain in Italy. It extends from the Alps in the north to the Apennines in the south-west, and the Adriatic Sea to the east. It comprises the Po river basin, the eastern lowlands of Veneto and Friuli, and the south-eastern basins of Emilia-Romagna. The Po Valley is one of the most developed and populated areas in Italy, generating about half of the country's gross domestic product (AdBPo, 2006). In the lower part of the Po river, flood-prone areas are protected by a complex system of embankments and hydraulic works that are part of the flood defence system in the Po Valley, extending for almost 3,000 km as a result of centuries-long tradition of river embanking (Govi and Turitto, 2000; Lastoria et al., 2006; Masoero et al., 2013). However, flood protection structures generate a false sense of safety and low risk awareness among the floodplain residents (Tobin, 1995). As a result, exposure has steadily increased in flood prone areas of the Po Valley (Domeneghetti et al., 2015). Records of past flood events (1950-1995) maintained by the National Research Council (Cipolla et al., 1998) show that more than 3,300 individual locations were affected by approximately 1,000 flood events within the Po Valley.





Three of the most recent flood events within the Po Valley (figure 1) have been chosen as case studies for this
analysis: the 2002 Adda flood that affected the province of Lodi (**1**); the 2010 Bacchiglione flood which
involved the area of Vicenza (**2**); and the 2014 Secchia flood in the province of Modena (**3**). All three locations
have been subject to frequent flooding between 1950 and 2000 according to the historical catalogue. A short
description of these three events is provided hereinafter to understand the dynamics and the impacts of each
flood.

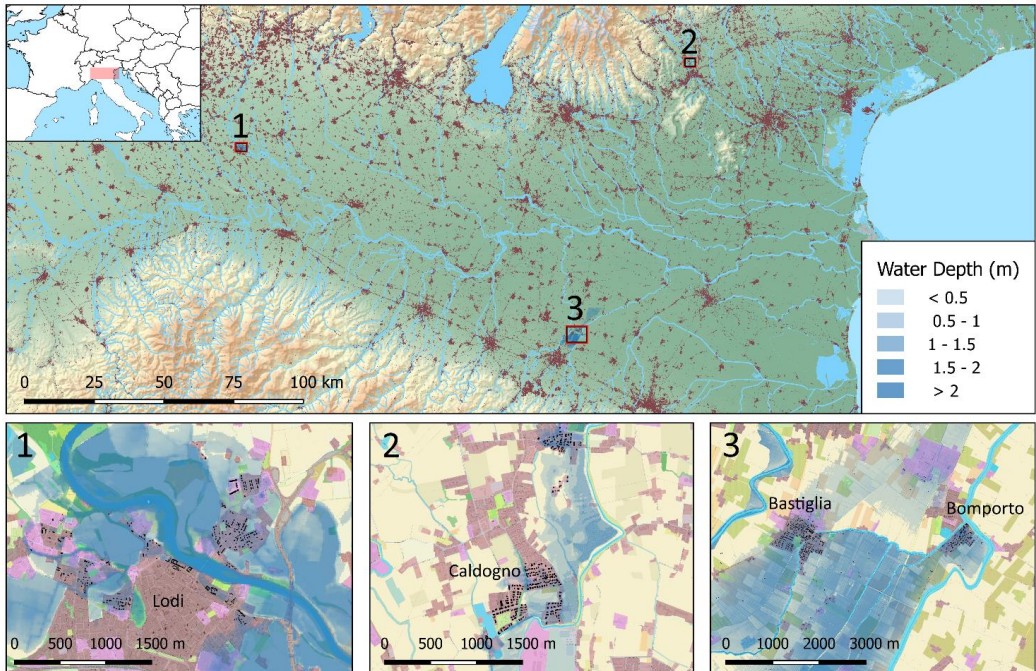

**Figure 1.** Case studies in Northern Italy (Po Valley). 1: Adda river flooding the town of Lodi, 2002; 2: Bacchiglione river
flooding the province of Vicenza, 2010; 3: Secchia river flooding the province of Modena, 2014. Flooded buildings for
which damage records are available are shown in black.
*2.1.1   Adda 2002*
On the 27th of November 2002, the province of Lodi (Lombardy) was struck by a flood caused by the
overflow of the Adda river. The flood-wave reached a record discharge of about 2,000 m³/s, corresponding to
a return period of 100 years (Rossetti et al., 2010). The river overtopped the embankments and flooded the
rural area first, later reaching the residential and commercial areas within the capital town of the province,
Lodi. The low-speed flood waters rose up to 2.5-3m. The inundation lasted for about 24 hours and affected a
large area of the Adda floodplain, including 5.5 ha of residential buildings. There were no reported
casualties, but several families were evacuated during the emergency and important service nodes such as
hospitals were severely affected. The reported damage to residential properties, commercial assets and
agriculture summed up to EUR 17.7M, out of which EUR 7.8M relate to residential buildings.



### 2.1.2 *Bacchiglione 2010*

From the 31st of October to the 2nd of November 2010, persistent rainfall affected the pre-Alpine and foothill areas of Veneto region exceeding 500 mm in some locations (ARPAV, 2010). As a result, about 140 km² of land were flooded, involving 130 municipalities (Belcaro et al., 2011). The Bacchiglione river, in the province of Vicenza, was particularly negatively affected. Heavy precipitation events and early snow melting increased the hydrometric levels of the Bacchiglione river and its tributaries, surpassing historical records (Belcaro et al., 2011). On the morning of November 1st, the water flowing at 330 m³/s opened a breach on the right levee of the river, flooding the countryside and the settlements of Caldogno, Cresole and Rettorgole with an average water depth of 0.5 m (ARPAV, 2010). Then the river overflowed downstream, towards the chief-town of Vicenza, which was inundated up to its historical center. The inundation lasted for about 48 hours, and its extent was about 33 Ha, of which 26 Ha consisted of agricultural land and 7 Ha were urban areas. The total damage including residential properties, economic activities, agriculture and public infrastructures was estimated to be around EUR 26M, while dwellings alone accounted for EUR 7.5 M (Scorzini and Frank, 2015).

### 2.1.3 *Secchia 2014*

In January 2014 severe rainfall endured for two weeks on the central part of Emilia-Romagna region, discharging the annual average amount of rain in just a few days. On the 19th, at around 6 AM, the water started to overtop a section (10 m) of the of the right levee of the Secchia river, which stands 7-8 meters over the flood plain. Later in the morning the levee breached at the top by one meter, flooding the countryside. After 9 hours, the levee section was completely destroyed for a length of 80 meters, spilling 200 m³/s and flooding around six thousand hectares of rural land (D'Alpaos et al., 2014). Seven municipalities were affected, with the small towns of Bastiglia and Bomporto suffering the largest impact. Both towns, including their industrial districts, remained flooded for more than 48 hours. The total volume of water inundating the area was estimated to be around 36 million m³, with an average water depth of 1 meter (D'Alpaos et al., 2014). The economic cost inflicted to residential properties according to damage declaration amounts to EUR 36M.

## 3. Materials and methods

### 3.1 Data description

The dataset we compiled for this analysis comprises:

- Detailed hazard data, including the flood extent, depth, persistence, and flow velocity.
- High-resolution spatial exposure data, including type, location and value of affected buildings.





▪  Comprehensive vulnerability data, including the characteristics of building and dwellings in terms
2        of material, quality and age.
▪  Reported costs of reparation or replacement of damaged goods.
The main hazard features (extent, depth, flow velocity and duration) are obtained from flood maps
produced by 2D hydraulic models based on observations performed during and after the events. In detail,
the hydraulic simulation for the Adda river has been produced by means of River2D model (Steffler and
Blackburn, 2002) using a 5m resolution digital terrain model and high-resolution LiDAR data for the
description of the floodplains obtained from the river basin district authority. The Bacchiglione flood have
been simulated using the 1D/2D model Infoworks RS (Beta Studio, 2012). The 1D river network geometry
comes from a topographic survey of cross-sections, while the 2D floodplain morphology (5 m resolution) is
obtained from LiDAR data produced by the Italian Ministry of Environment (Molinari et al., 2018). The
reliability of the simulations for the Adda and Bacchiglione floods was verified using hydrometric data,
aerial surveys of inundated areas and photos/videos from the affected population (Rossetti et al., 2010;
Scorzini and Frank, 2015). The Secchia flood event has been simulated using an innovative, time-efficient
approach (Vacondio et al., 2016) which integrates both river discharge and floodplain characteristics in a
parallel computation. The simulation was performed on a 5 meters grid and its results were validated
against several field data and observations, including a high-resolution radar image (Vacondio et al., 2014,
2017). The information needed for the characterization of exposure is collected from a variety of sources and
then spatially projected to have a homogeneous, georeferenced dataset for each case study. External
buildings perimeter and area are obtained from the Open Street Map database (Geofabrik GmbH, 2018) and
associated with official street-number points containing addresses. The land cover is freely available as
perimeters classified by the CORINE legend (4th level of detail) (Feranec, J. Ot'ahel', 1998) obtained from
Regional Environmental Agencies databases. Land cover information is used to discriminate housing from
other buildings (industrial, commercial, etc.). In addition, indicators for building characteristics (Table 1)
have been selected from the database of the official Italian Census of 2011 (ISTAT). Construction and
restoration costs as $EUR/m^2$ are obtained for the case study areas from the CRESME database
(CRESME/CINEAS/ANIA, 2014). They are used to convert the absolute damage values into relative damage
shares. Empirical damage records have been collected by local administrations after the flood events in
relation to households' street numbers. The records falling outside the simulated flood extents are filtered
from the dataset. Each record includes: claimed; verified; and refunded damage to residential buildings.
Since actual compensation often covers only a fraction of the damage costs, claimed damage is preferred in
order to measure the economic impact (see Carisi et al. 2018). We restricted our analysis on direct monetary
damage to the structure of residential buildings, excluding furniture and vehicles. Economic losses, building
values and construction costs for the three events have been scaled to EUR2015 inflation value.



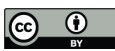

## 3.2 Damage models overview

Empirical damage models are drawn based on actual data collected from specific events (e.g. Luino et al. 2009; Hasanzadeh Nafari et al. 2017); in some regions they represent the only knoweldge base for the assessment of flood risk. However, they carry a large uncertainty when employed in different times and places (Gissing and Blong, 2004; McBean et al., 1986). Differently, synthetic models are based on a valuation survey which assesses how the structural components are distributed in the height of a building (Barton et al., 2003; Oliveri and Santoro, 2000; Smith, 1994). In such expert-based models, the magnitude of potential flood loss is estimated based on the vulnerability of structural components via "what-if" analysis and in the evaluation of the corresponding damage based on building and hazard features (Gissing and Blong, 2004; Merz et al., 2010). Most empirical and synthetic models are UVMs based on water depth as the only predictor of damage; yet recent studies (see e.g. Dottori et al., 2016 and Merz et al. 2013) suggest that MVMs developed using expert-based or machine-learning approaches outmatch the performances of customary univariable regression models. However, the development of MVMs requires a comprehensive set of data in order to correctly identify complex relationships among variables.

### 3.2.1 Models from literature

There are few models in the literature that are dedicated to the economic assessment of flood impacts over Italian residential structures (see e.g. Oliveri and Santoro 2000; Huizinga 2007; Luino et al. 2009; Dottori et al. 2016). These models have been developed independently using different approaches, assumptions and base data. The first model we selected for testing (Luino et al., 2009) is an empirical UVM based on the impact data collected after the flash-flood event of May 2002 in the Boesio Basin, in Lombardy. One stage-damage curve was generated for structural damage to the most common building type in the area using loss data measured after the flood combined with estimates of water depth from a 1D hydraulic model. In this model, the estimation of building value is based on its geographical location, use and typology, based on market value quotations by the official real estate observatory of Italy (Agenzia delle Entrate, 2018). Market values of residential stocks for specific areas. The second model (OS - Oliveri and Santoro 2000) is a synthetic UVM developed for a study performed in the city of Palermo (Sicily). The model is based on water depth and consists of two curves, one for buildings with 2 floors and one for those with more than 2 floors. It considers water stage steps of 0.25 m; for each stage, the model computes the overall replacement cost as the result of damage over different components (internal and external plaster, fixtures, floors and electric appliances) plus the expenses for dismantling the damaged components. The third model we included in our analysis is part of a stage-damage curve database developed by the EU Joint Research Centre (Huizinga, 2007; Huizinga et al., 2017) on the basis of an extensive literature survey. Damage curves are provided for a variety of assets and land use classes on the global scale by normalising the maximum damage values in relation to country-specific construction costs. These are obtained by means of statistical regressions with socioeconomic



development indicators. The JRC curves are suggested for application at the supra-national scale but can be
a general guide to carry on assessments at the meso-scale in countries where specific risk models are not
available. We select the curve provided for Italy (JRC-IT) to be tested on our dataset.

The fourth model considered is INSYDE, *In-depth Synthetic Model for Flood Damage Estimation* (Dottori et al. 2016), which is a synthetic MVM developed for residential buildings and released as open source R script. Repair or replacement costs are modelled by means of analytical functions describing the damage processes for each component as a function of hazard and building characteristics, using an expert-based "what-if" approach. Hazard features include physical variables describing the flood event at the building location, e.g. water depth, flood duration, presence of contaminants and sediment load.

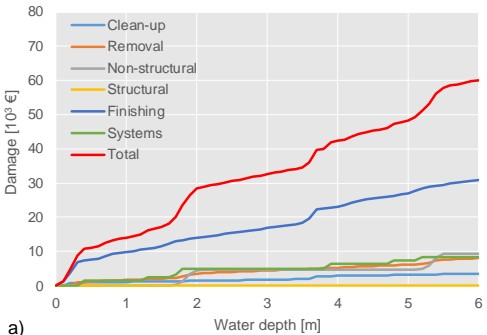

**Figure 2.** Examples of damage curves in relation to water depth produced by INSYDE for riverine floods in relation to a building with FA=100 m², NF=2, BT=3, BS=2, FL=1, YY=1990, CS=1.

Exposure indicators include building characteristics such geometry and features. Building features affect
costs estimation either by modifying the damage functions or by affecting the unit prices of the building
components by a certain factor. Damage categories include clean-up and removal costs, damage to finishing
elements, windows, doors, wirings and installations (Figure 2). The model adopts probabilistic functions for
some of the buildings' components for which it is difficult to define a deterministic threshold of damage
occurrence in relation to hazard parameters. The list of explicit input variables accounted by INSYDE is
shown in Table 1, with the indication of their respective data sources. Despite the large number of inputs, the
model proved to be adaptable to the actual available knowledge of the flood event and building
characteristics (Molinari and Scorzini, 2017).

| Variable | Description | Source | Unit | Name |
|---|---|---|---|---|
| **Hazard features** | | | | |
| Water depth | Maximum depth | Hydro model | m | he |
| Flow velocity | Maximum velocity | Hydro model | m/s | v |
| Duration | Hours of inundation | Hydro model | h | d |
| **Exposure and vulnerability of buildings** | | | | |
| Replacement value | Economic value of the building structure | CRESME | EUR/m² | RV |
| Area and perimeter | Footprint area and external perimeter | OSM/CTR | m², m | FA, EP |
| Basement | Presence (1) or absence (0) of basement | CRESME | - | B |
| Number of floors | 1, 2, 3 or more than 3 floors | Census/Inspection | - | FN |
| Building type | Flat (1), semi-detached (2) or detached (3) | Census/Inspection | - | BT |
| Building structure | Bricks (1) or concrete (2) | Census/Inspection | - | BS |
| Finishing level | Low (0.8), medium (1) or high (1.2) | Census/Inspection | - | FL |



| Conservation status | Bad (0.9), normal (1) or good (1.1) | Census/Inspection | - | CS |
|---|---|---|---|---|
| Observed damage | | | | |
| Damage claims | Private and shared structural parts | Official survey | EUR | D |

**Table 1.** List of variables included in the multivariable analysis.
*3.2.2    Models developed and trained on the observation dataset*
This section provides an overview about the empirical damage model obtained from our events dataset,
namely two supervised learning algorithms (Random Forest, Artificial Neural Network) and three uni- and
bivariable regression models used to assess the importance of variables as damage predictors. All these
models share the same sampling approach for training and validation: the observation dataset is split in
three parts, where two thirds are used to train the model and one third for validation.
*3.2.2.1    Multivariable models: supervised learning algorithms*
A probabilistic approach is required in damage estimation in order to control the effects of data variability
on the model uncertainty. This is useful to overcome the limitations associated with the choice of a singular
model and to increase the statistical value of the analysis (Kreibich et al., 2017). The algorithms we employed
to deal with the empirical data share an iterative scrambling and resampling approach (1,000 repetitions) in
order to draw the confidence interval of the models independently from source data variability. For the
setup of empirically-based MVMs we selected ten variables from those listed in Table 1, excluding those
with small variability (basement, conservation status) or those for which an adequate level of detail is not
possible in our case studies (age, heat system). These ten variables serve as input for two machine learning
algorithms, namely Random Forest (RF) and Artificial Neural Network (ANN), described in the next
paragraphs. Both algorithms produce a distribution of estimates for each record, from which the mean value
is calculated.
Random Forest
The RF is a data mining procedure, a tree-building algorithm that can be used for classification and
regression of continuous dependent variables (CART method - see Breiman 1984) like the one used
by Merz et al. (2013). RF has numerous advantages, such high prediction accuracy, tolerance of
outliers and noise, avoidance of overfitting problems, and no need of assumptions about
independence, distribution or residual characteristics. Because of this, it has already been
employed in the context of natural hazards, including earthquake-induced damage classification
(Tesfamariam and Liu, 2010), flood hazard assessment (Wang et al., 2015), and flood risk (Carisi et
al., 2018; Chinh et al., 2015; Kreibich et al., 2017; Merz et al., 2013; Spekkers et al., 2014).



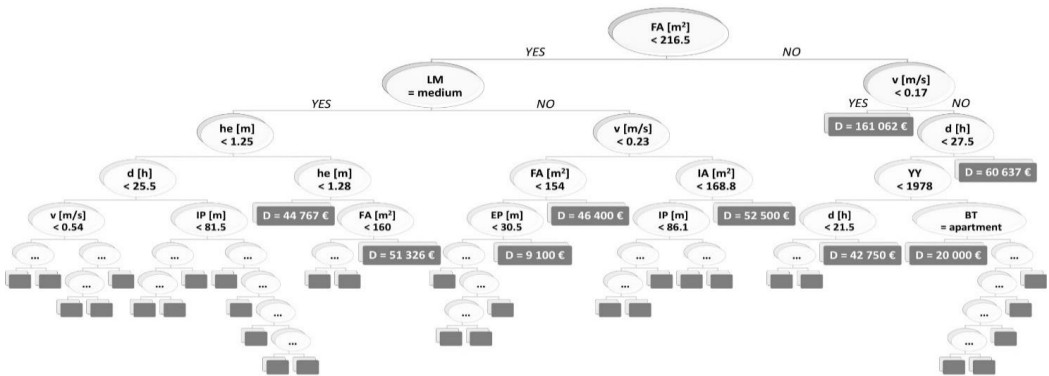

**Figure 3.** Example of one of the regression trees produced by the Random Forest model.

We use the algorithm implemented in the R package *RandomForest* by Liaw and Wiener (2002). The Random Forest algorithm builds and combines many decision trees, where each tree has a non-linear regression structure, recursively splitting the input dataset into smaller parts by identifying the variables and their splitting values which maximize the predictive accuracy of the model. The tree structure has several branches, each one starts from the root node and includes several leaf nodes, until either a threshold for the minimum number of data points in leaf nodes is reached or no further splitting is possible. Each estimated value represented by the resulting terminal node of the tree answers to the partition question asked in the previous interior nodes and depends on the response variable of all the parts of the original dataset that are needed to reach the terminal node (Merz et al., 2013). In order to reduce the uncertainty associated with the selection of a single tree, the RF algorithm (Breiman, 2001) creates several bootstrap replicas of the learning data and grows regression trees for each subsample, considering a limited number of variables at each split. This will result in a great number of regression trees, each based on a different (although similar) set of damage records and each leaving out a different number of variables at each split. The mean value among all prediction of the individual trees is chosen as representative output. An example of a built tree for the present study is shown in Figure 3. Another important strength of RF is its capability to evaluate the relative importance of each independent variable in the tree-building procedure, i.e., in our case, in representing the damage process. By randomly simulating the absence of one predictor, the RF algorithm calculates the decreasing of the performance of the model and thus the importance of the variables in the prediction.

Artificial Neural Network

ANNs are mathematical models based on non-linear, parallel data processing (Haykin, 2001). They have been applied in several fields of research, such as hydrology, remote sensing, and image classification (Campolo et al., 2003; Giacinto and Roli, 2001; Heermann and Khazenie, 1992). The model used in this study (Essenfelder, 2017) consists of a Multi-Layer Perceptron (MLP) neural network model, using back-



propagation as the supervised training technique and the Levenberg-Marquardt as the optimization
algorithm (Hagan and Menhaj, 1994; Yu and Wilamowski, 2011) (see figure 4 for the structure of the model).

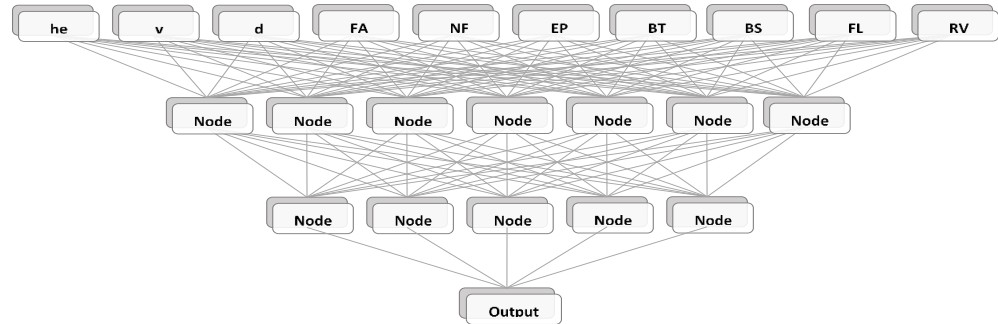

**Figure 4.** Structure of the Artificial Neural Network model applied in this study using two neurons (nodes) layers.
The developed ANN model evaluates the Sum of Squared Errors (SSE) of the model outputs with regards to
the targets for each training epoch as a way of assessing the generalization property of a trained ANN model
(Hsieh and Tang, 1998; Maier and Dandy, 2000). The ANN runs in a multi-core configuration and provides
an ensemble of trained ANN models as a result, thus being suitable for probabilistic analysis. The input and
target information are normalized by feature scaling before being processed by the model, while the initial
number of hidden neurons per hidden layer is approximated as two-thirds of the summation of the number
of neurons in the previous and next layers (Han, 2002). Regarding the activation functions, a log-sigmoid
function is used for the connection with neurons in the first and second hidden layers, while a linear
function is used for the connections with neurons in the output layer, allowing values to be either lower or
greater than the maximum observed valued in the target dataset. This configuration is interesting as it does
not limit the output range of the ANN model to the range of normalized values. The input data is randomly
split between three distinct sets, namely training, validation, and test. The training dataset is used to
calibrate the ANN model, meaning that the weight connections between neurons are updated with respect
to the data available in this dataset. The validation set is utilized to avoid the overtraining or overfitting of
the ANN model, being used to stop the training process. The test set is not presented to the model during
the training procedure, being used only as a way of verifying the efficiency of a trained ANN when stressed
by new data. In order avoid any possible bias coming from the random split of the original dataset into
training, validation, and test datasets, about 1,000 training attempts are performed, each with a different
initial weight initialization and training dataset composition. The resulting ANN model consists of an
ensemble of 4 models, representing the best overall results after the training procedure, that are used to
define the confidence interval.



*3.2.2.2    Univariable and bivariable models*
In order to understand if the added complexity of MVMs brings any improvement in the accuracy of
damage estimates, we compare them with traditional, deterministic univariable (UVM) and bivariable
(BVM) regression models that are empirically derived from the observation dataset. Considering the first
(water depth) or the first two variables (water depth and water velocity), we investigate whether a linear,
logarithmic or exponential function has the best regression fit to the records. All functions that consider
water depth are forced to pass through the origin, because most buildings have no basement and,
accordingly, no water means no damage. Similarly to what we did for the MVM training, we uses an
iteration of 1,000 scrambling and resampling cycles which is repeated using the two different sampling
strategy: first the models are trained on 2/3 of the data and validated on the remaining 1/3 of the records.
## 4.    Results and discussion
### 4.1    Observed damage records
Our combined dataset contains records of 1,158 damaged residential buildings (Table 2). More than a half of
these were damaged by the Secchia flood, which affected the largest residential area (17.7 ha) and caused the
largest total losses. Only verified, spatially-matching records are accounted; economic losses are scaled to
EUR2015 inflation value. Note that these losses are related to the structural damage of residential buildings,
thus they do not represent the full cost of the events.

| Case study [River basin, year] | Affected buildings [n] | Flood extent [ha] | Avg. water depth [m] | Declared damage [M EUR 2015] |
|---|---|---|---|---|
| **Adda**, 2002 | 270 | 5.5 | 0.8 | 4.7 |
| **Bacchiglione**, 2010 | 294 | 7.1 | 0.5 | 7.9 |
| **Secchia**, 2014 | 594 | 17.7 | 1 | 21.1 |
| **Total** | 1,158 | 30.3 | 2.3 | 33.7 |

**Table 2.** Summary of residential buildings affected by the three investigated flood events according to hydraulic
simulations and damage claims.
Boxplots in Figure 5 show the variance of variables driving the damage. Water depths range from 0.01 to
about 2 meters, with most records falling in the interval 0.4 – 1.2 meters. Water velocities range between 0.01
and 1.5 m/s. Footprint areas and observed relative damages have similar average values for all three events,
however the Secchia case study presents the longer count of records as well as the largest spread of outliers.

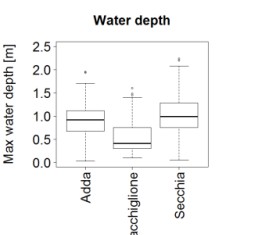
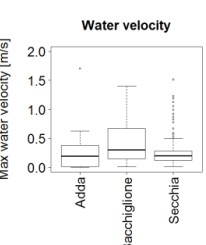
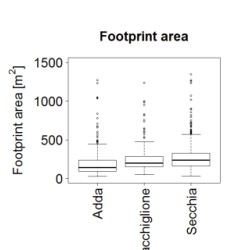
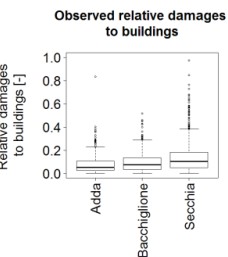



**Figure 5.** Data distribution for four variables from the three sample case studies.
The scatterplot in Figure 6 better shows the density of observed damages records in relation to the maximum
water depth. The increase in depth corresponds to a larger range of variability in the economic damage.

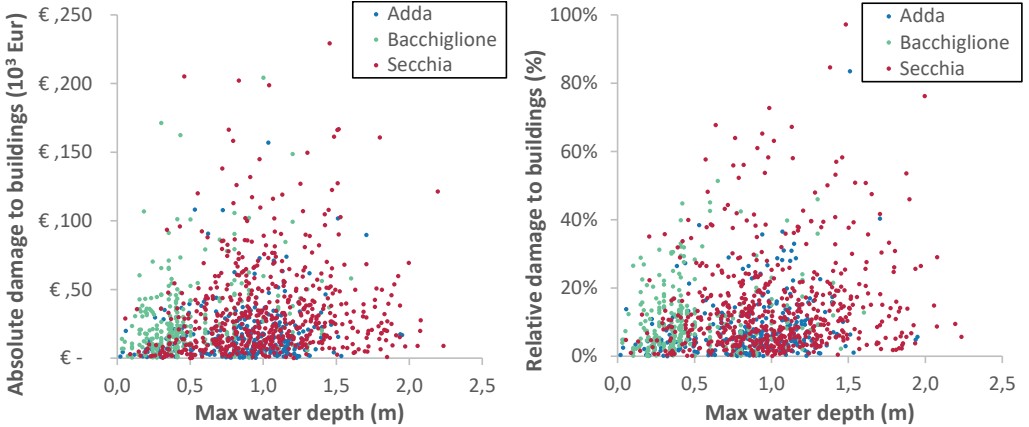

**Figure 6.** Scatterplot of monetary **(left)** and relative **(right)** damage (y-axis) in relation to maximum observed water
depth (x-axis). Records from the same event are shown with the same color.
### 4.2    Influence of hazard and exposure variables on predicting flood damage

Water depth (*he*) is identified by RF as the most important predictor of damage (factor 3.4) among the ten examined variables (Figure 7). This confirms previous findings (Wagenaar et al., 2017b) and justifies the use of depth-damage curves for post-disaster need assessment. Flow velocity and geometric characteristics of buildings (area and perimeters) are also important (factor 2.7 to 2.3), followed by other predictors such as building value, flood duration, number of floors, finishing level and type of structure (factor 1 or less). Although water depth is the most influential variable, it is only moderately more important than other predictors. That substantiates the efforts to test the applicability of multivariable approaches to improve the estimation of damage.

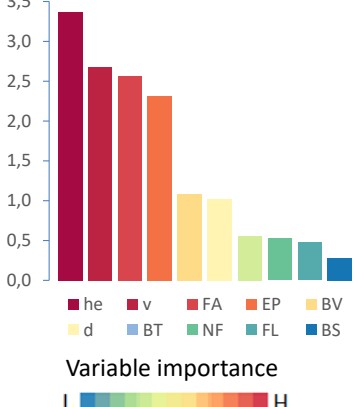

**Figure 7.** Relative importance of variables as predictors of damage according to the RF model.

### 4.3    Performance of the models
For assessing the predictive capacity of the four selected literature models, we compare them with
empirically-based, data-trained models structured on the same variables, i.e. the evaluation of the models'
performances is carried out by measuring and comparing the error metrics from the aforementioned models



(JRC-IT, Luino, OS and INSYDE) to those provided by the empirical MVMs obtained from supervised
learning algorithms, the BVMs and traditional UVMs (depth-damage curves) developed on our dataset. The
performances of each model are evaluated by using three metrics, namely Mean Absolute Error, Mean Bias
Error and Root Mean Square Error. The MAE indicates the precision of the model in replicating the total
recorded damage. The MBE shows the systematic error of the model, which is its mean accuracy. The RMSE
measures the average magnitude of the error, enhancing the weight of larger errors. In addition to these
error metrics, the total percentage error (E%, difference between observed and simulated damage divided by
observed damage) is shown in tables.
*4.3.1 Literature models*
As first step, estimates of empirical and synthetic models from literature are compared with observed
damages and the results in terms of total loss and total percentage error are shown in Table 3.

| Case study | Unit | Obs. | JRC-IT | LUINO | OS | INSYDE |
|---|---|---|---|---|---|---|
| **Adda** | M EUR 2015 | 4.7 | 24.3 | 13.0 | 8.1 | 5.6 |
| **2002** | E% | | 417.0 | 176.6 | 72.3 | 19.1 |
| **Bacchiglione** | M EUR 2015 | 7.9 | 19.2 | 11.4 | 6.5 | 8.3 |
| **2010** | E% | | 143.0 | 44.3 | -17.7 | 5.1 |
| **Secchia** | M EUR 2015 | 21.1 | 64.5 | 44.1 | 19.8 | 28.8 |
| **2014** | E% | | 205.7 | 109.0 | -6.2 | 36.5 |
| **Full set** | M EUR 2015 | 33.7 | 108.0 | 68.5 | 34.4 | 42.7 |
| | E% | | 220.5 | 103.2 | 2.0 | 26.7 |

**Table 3.** Estimates and error from literature models compared to observed damage. Monetary values are in Million Eur,
E% is total percentage error.
JRC-IT is the worst performing model, largely overestimating damage from the three events (E% 143-417),
followed by the UV empirical model from Luino which overestimates damage with a percentage error
ranging from 44 to 177. These results indicate that meso-scale models are not suitable for application at the
micro-scale and that empirical models should be carefully applied for flood events with different
characteristics from the ones for which they are developed. In fact, Luino's model was produced for a flash-
flood event, with higher velocities and impacts. The two synthetic models, OS and INSYDE, perform much
better, yet showing a large variability of the error factor, depending on the considered case. In detail, OS
provides better results for the Secchia event (6% underestimation) and worse for the Adda set (72%
overestimation), resulting in an estimate that is very close to the observations in terms of percentual error on
the total dataset, although this is mainly due to compensation of positive and negative errors for the
different events. Differently, the INSYDE model exhibits a better performance for the Bacchiglione event (5%
overestimation) and worse for the Secchia case study (37% overestimation). It is worth noting that, although
the accuracy of the OS model is higher than of the INSYDE model for the full set, the latter is more accurate
for two out of the three case studies (i.e. Adda 2002 and Bacchiglione 2010). Moreover, the INSYDE model





provides more precise results, with a variance in errors 10 times lower than of the OS model and with
maximum errors never exceeding an absolute value of 40%. However, INSYDE seems to consistently
overestimate the total damages. Figure 8 compares the estimated and observed damages for the entire
dataset for the two best performing literature models (OS and INSYDE).

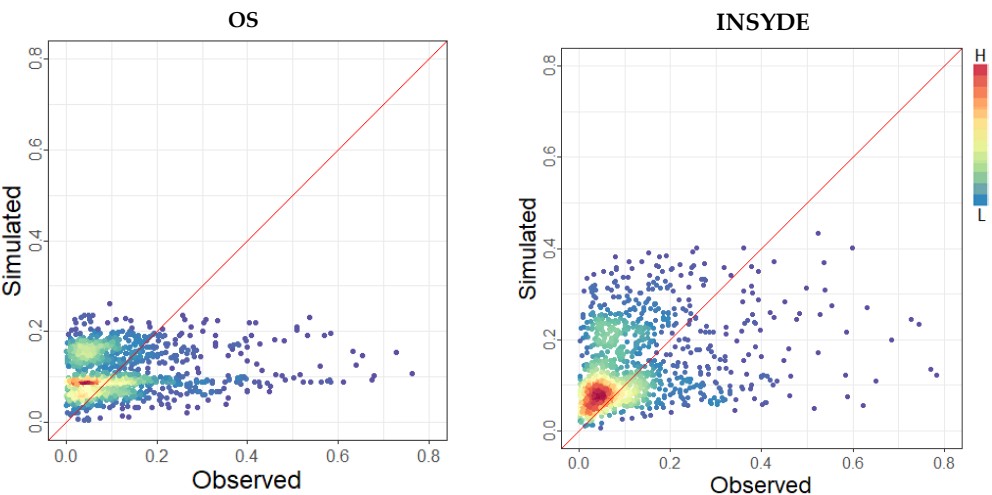

**Figure 8.** Scatterplot comparing relative damage estimates produced by the two best performing literature models, OS **(left)** and INSYDE **(right).** Simulated damage on the y-axis, observed damage on the x-axis. Colors represent records density.

*4.3.2   Data-trained univariable, bivariable and multivariable models*
In this section, damage values estimated by empirical, data-trained UVMs, BVMs and MVMs are compared
with observed damage data. The results provided by these empirically-based models are used as a
benchmark to understand the capability of tested literature models in predicting damage. The error metrics
chosen for comparing the models' performances are presented for relative damage based on official
estimates of replacement value, however training and validation were carried out also in terms of monetary
damage with similar results, not presented for the sake of brevity.

| Function | UVMs | | | BVMs | | |
|---|---|---|---|---|---|---|
| | MBE | MAE | RMSE | MBE | MAE | RMSE |
| **Linear** | -0.015 | 0.087 | 0.127 | -0.012 | 0.087 | 0.126 |
| **Log** | -0.046 | 0.080 | 0.131 | -0.046 | 0.080 | 0.131 |
| **Root** | -0.003 | 0.086 | 0.123 | -0.002 | 0.086 | 0.123 |

**Table 4.** Error metrics for the Univariable and Bivariable models.

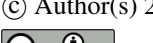
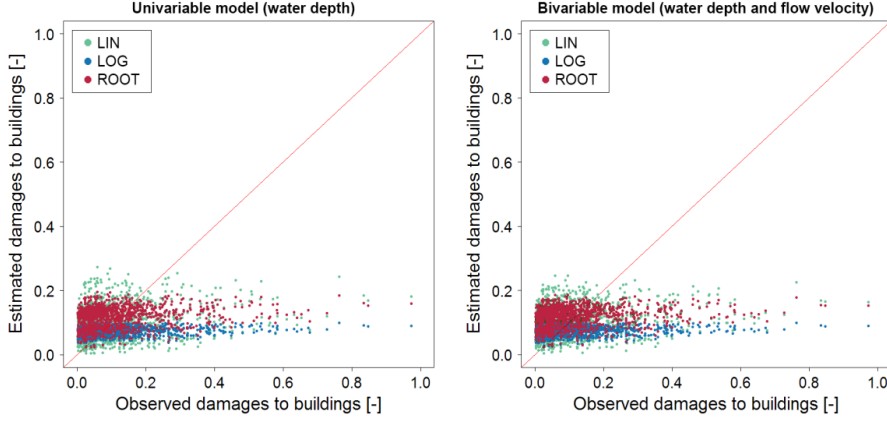

**Figure 9.** Testing the predictive capacity of uni- and bivariable models: estimated relative damage (y-axis) from the UVM
**(left)** and BVM **(right)** are plotted against observed relative damage (x-axis) according to the three tested regression
functions (LINear, LOGarithmic and ROOT function).
The results shown in Table 4 and figure 9 indicate no significant differences between UVMs and BVMs. We
can affirm that the inclusion of water flow velocity as complementary explanatory variable does not improve
the performance of simple regression models in our case study. For this reason, BVMs are dropped from
further discussion from now on, to focus on a direct comparison between UVMs and MVMs.
Taking into consideration only UVMs, MAE and RMSE are very similar for the three tested regression
functions. However, the root function described by the general formula $y = b(\sqrt[q]{x})$ has a slightly better fit
(correlation is higher, MBE is lower) compared to linear and log functions. We select the function described
by the equation $y = 0.13(\sqrt{x})$ as the best performing UVM to be included in the comparison with MVMs.
Our findings confirm previous results indicating that the root curve as the most adequate to describe the
flood damage process (Buck and Merkel, 1999; Cammerer et al., 2013; Elmer et al., 2010; Kreibich and
Thieken, 2008; Penning-Rowsell et al., 2005; Scawthorn et al., 2006; Sluijs et al., 2000; Thieken et al., 2008;
Wagenaar et al., 2017b).
Figure 9 shows a direct comparison between the damage estimated by the empirically-based models against
observed damage. The upper panel shows the output from the UVM described by the root function. The
lower panels show the output of the RF (left) and ANN (right) algorithms. The two machine learning
algorithms produce comparable results, with both RF and ANN models tending to slightly overestimate the
average damage (higher density of points, in red) and to significantly underestimate extreme values (lower
density of values, in blue). This is a common result of data-driven models, where better results are biased to
high-frequency values in comparison to low-frequency values due to the larger sample of those data in the
calibration dataset. In Figure 10, the range of estimates, shown as min-max, describes the confidence of the
model for individual records. In the case of RF, it shows the min-max range over all the 1,000 iterations of

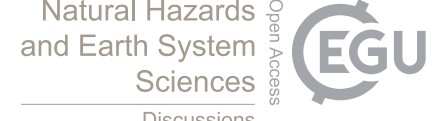

the model, while in the case of ANN only an ensemble of the four best-fit models is shown (see Section

2    3.2.2.1).

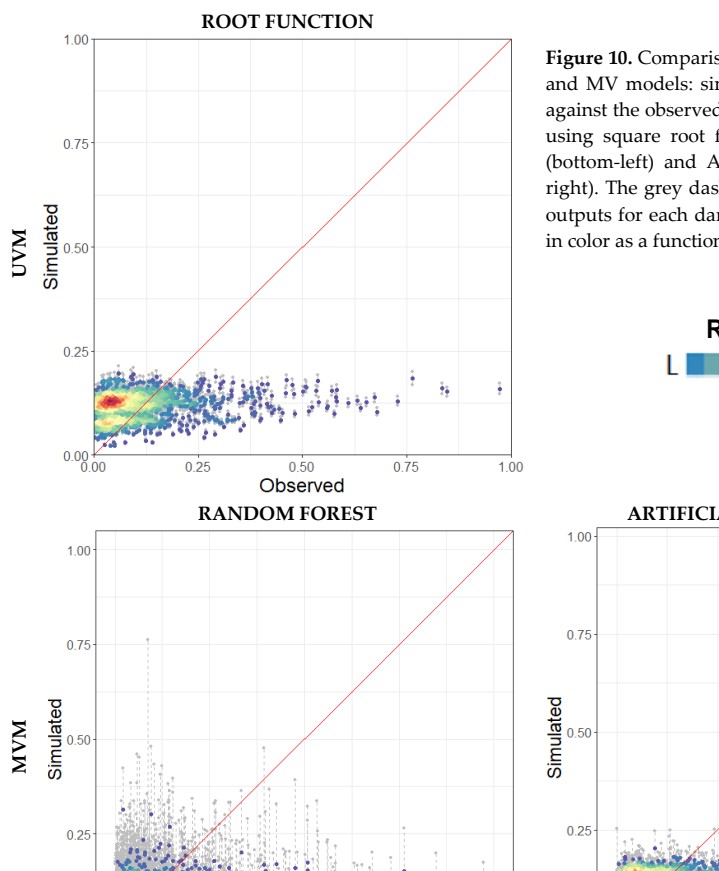

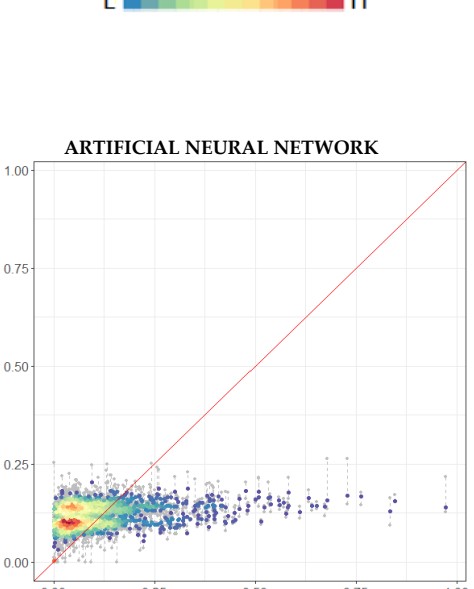

**Figure 10.** Comparison of the predictive capacity of UV and MV models: simulated damage (y-axis) is plotted against the observed damage (x-axis) for the UV model using square root function (top-left), Random Forest (bottom-left) and Artificial Neural Network (bottom-right). The grey dashed line shows the range of model outputs for each damage record. The median is shown in color as a function of the record density.

Theoretically, MVMs should simulate the complexity of the flooding mechanism better than UVMs. In our
test, the ANN model has the best fit to the data, but UVMs (depth-damage curves) appear to perform
similarly: the MVMs describe recorded damage with a percentage error between 0.2 and 10, while UVMs'
error is around 12 (see table 5 in the next paragraph). Accordingly, when extensive descriptive data are not
available, UVMs appear to be a reasonable alternative to describe the flood damage process. These
empirically data-driven models are useful to understand the capability of multivariable approaches in
predicting damage, i.e. which is the range of uncertainty that can be expected when assessing the flood
damage process, comparing to simpler models like UVMs.



### 1   *4.3.3   Comparing models' performances*

First, we evaluate how selected literature UVMs (JRC-IT, Luino and OS) compare to the root function trained
on the empirical dataset. Figure 10 shows the distribution and the density of observed relative damage as a
function of water depth for the full dataset, together with the UV curves selected for testing. This figure
explains the results presented in Section 4.3.1, with the JRC-IT and Luino models growing too fast for
shallow water depths, as opposed to OS (shown as two separate curves for different number of floors of the
building), which has a good mean fit to the data.

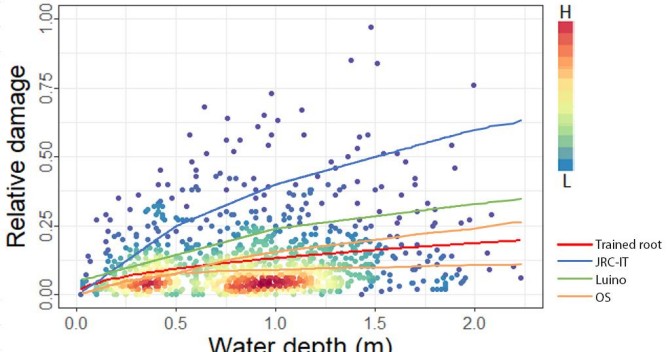

**Figure 11.** Scatterplot of relative damage records (y-axis) and water depth (x-axis). Points color represents record density. The red line shows the empirical root function ($y = 0.13(\sqrt{x})$, selected as best fit. The other lines represent the three UV literature models (JRC-IT, Luino, and OS) selected for the test. OS model is made of two curves, in relation to the number of floors of the building.

Table 5 summarises the main results from all the models in terms of error metrics. Specifically, among all
models, MVMs RF and ANN are those with the lowest MAE and RMSE, followed by UVM ROOT with a
MAE of 0.086 and a RMSE of 0.123. In terms of percentage error, the ranking is the same, with the only
exception of OS, whose result in terms of this metric lies between the two empirical data-trained MVMs.
Overall, the two expert-based literature models OS and INSYDE, are the best performing ones when
compared to empirically-trained models, as shown by MAE, MBE and RMSE. As mentioned before, the
performance of the UVM OS is very close to those of the MVM INSYDE, although this result may depend on
the fact that the large share of records come from the Secchia event, for which OS outperforms INSYDE.

| | Model | MBE | MAE | RMSE | Est. dmg [M EUR 2015] | Abs. error [M EUR 2015] | Percent error [%] |
|---|---|---|---|---|---|---|---|
| **Trained models** | **UVM (ROOT)** | -0.003 | 0.086 | 0.123 | 37.8 | +4.1 | +12.3 |
| | **MVM (RF)** | -0.024 | 0.081 | 0.126 | 30.4 | -3.3 | -9.8 |
| | **MVM (ANN)** | +0.009 | 0.091 | 0.115 | 33.8 | -0.1 | -0.2 |
| **Literature models** | **UVM (JRC_IT)** | +0.217 | 0.239 | 0.27 | 108 | +74.3 | +220.5 |
| | **UVM (Luino)** | +0.082 | 0.13 | 0.154 | 68.5 | +34.8 | +103.2 |
| | **UVM (OS)** | -0.009 | 0.088 | 0.127 | 34.4 | +0.8 | +2.0 |
| | **MVM (INSYDE)** | +0.019 | 0.093 | 0.132 | 42.7 | +9.0 | +26.7 |

**Table 5.** Comparing error metrics between empirically-base models and INSYDE.



Based on these results, the synthetic models INSYDE and OS currently represent very good alternatives for
flood risk assessment in Italy, in cases where no empirical loss data are available to develop specific damage
models. Indeed, our analysis has shown that particular care should be taken when transferring models
derived from specific events (Luino curve) or from different scales (JRC-IT), while synthetic models can be
considered more robust tools, relying on a physically-based description of flood damage mechanisms.
Overall, for the investigated dataset, the synthetic MVM INSYDE has not been found to provide an
improvement in the accuracy of damage estimates compared to those of the UV OS. However, the results of
INSYDE are more precise if considering the different flood events, with a general, although limited, damage
overestimation in all the cases, as opposed to OS which exhibited more accurate performance only for the
Secchia flood and larger variability for the other two events, consequently being less precise. Further
validation exercises, combined with the application of standardised and detailed procedures for damage
data collection (e.g. Molinari et al. 2014) could improve INSYDE's predictive accuracy; being an open-source
model, it is possible to modify the damage functions for the different building components; for example, the
availability of datasets with building losses subdivided into different categories (e.g. structural/non-
structural elements, finishing, systems, etc.) could help to identify which damage components are
responsible for the larger share of the error. The same cannot be said for OS, which is presented as a simple
stage-damage curve, without a detailed explanation of the modelling assumptions on the considered flood-
damage mechanisms.
As a final consideration, the accuracy and precision of damage observations are key aspects for the correct
development of an MVM. This makes synthetic and empirical MVMs better fit for applications at the micro-
scale (up to the census block scale (Molinari and Scorzini 2017)), where explanatory variables can be spatially
disaggregated. Indeed, the aggregation scale is of primary importance in the application of MVMs: if we can
compare our results to those reported in other studies applying similar multivariable approaches on an
extensive damage dataset (bagging of regression trees), as in Wagenaar et al. (2017a) and in Kreibich et al
(2017), we observe that our range of uncertainty is drastically smaller. This difference is likely imputable to
the fact that, in the referred studies, information is aggregated at the municipality level, as opposed to our
case, where each variable is precisely linked to buildings' location.
## 5.  Conclusions
Risk management requires a reliable assessment tool to identify priorities in risk mitigation and adaptation.
Such tool should be able to describe potential damage based on the available data related to hazard features
and exposure characterisation. Recent studies suggest that multivariable flood damage modelling can
outperform customary univariable models (depth-damage functions). In this study we collected a large
empirical dataset which includes multiple hazard and exposure variables for three riverine flood events in





Northern Italy, including the declared economic damage to residential buildings. On this basis, we produced
three univariable, three bivariable and two multivariable models that are compared in terms of predictive
accuracy and precision. We found that water depth is the most important predictor of flood damage,
followed by secondary variables related to hazard (flow velocity, duration) and exposure features (area,
perimeter and replacement value of the building). However, our results suggest that the inclusion of one
additional variable (flow velocity) does not improve the estimates produced by simple regression models in
a bivariable setup. On the other side, the analysis confirms the literature notion that the root function is the
best fitting curve to describe damage in relation to water depth. Two MVMs were trained using two
different machine learning algorithms, namely Random Forest and Artificial Neural Network. These
empirically-trained MVMs performed well (with an error ranging from 1 to 10%) in reproducing the damage
output from the three events and thus were set as a reference for assessments in the same geographic
context. In this perspective, other case studies are needed to confirm their robustness. Moreover, our results
corroborate previous findings about the advantages of supervised machine learning approaches for
developing or improving flood damage models. Still, their application remains limited by the availability of
the data required for the MVM setup. In case of scarce number of variables, however, simple univariable
models trained on the specific contexts seem to be a good alternative to MVMs.
We then considered four literature models of different nature and complexity to be tested on our extended
case study dataset. We compared their error metrics with those of the empirically-trained UVMs and MVMs
in order to evaluate their performance as predictive tool for flood risk assessment. The results have shown
that both UV (Oliveri and Santoro 2000) and MV (INSYDE, Dottori et al. 2016) synthetic models can provide
similar (although obviously larger) errors to those observed from empirical models. On the contrary, we
found important errors when transferring models derived from other specific events (Luino curve) or
different scales (JRC-IT). Therefore, the tested synthetic models can be currently considered as the best
option for damage prediction purposes in the Italian context, in cases where no extensive loss data are
available to derive a location-specific flood damage model. Overall, we found that errors produced by
synthetic models were smaller than 30% of observed damage, with INSYDE providing more precise results
over the different, single case study events (with a percentage overestimation of 19, 5 and 37% for Adda,
Bacchiglione and Secchia, respectively) and is more accurate for two out of the three case studies (i.e. Adda
and Bacchiglione), while the OS model is generally less precise but more accurate for the Secchia flood event
only (2% error, as opposed to a 72% overestimation for the Adda and 18% underestimation for the
Bacchiglione event).
Observed errors depend in part on the inherent larger variability found in the dataset related to that
particular event. Nevertheless, the collection of additional independent flood records from different
geographic contexts in Italy would help to further evaluate the adaptability of the models, especially of the





open-source INSYDE, to estimate their uncertainty, and to increase their predictive accuracy. Finally, the
work presented here has assembled a dataset that is currently one of the most extended and advanced for
Italy; on this track, we aim to promote a shared effort towards an updated catalogue of floods that includes
hazard, exposure and damage information at the micro-scale. To this purpose, the adoption of a
standardised and detailed procedure for damage data collection is a mandatory step.
**Data availability**
The INSYDE model is available as R open source code from https://github.com/ruipcfig/insyde
The hazard simulation of the Secchia flood event was kindly provided by Ing. Vacondio (University of
Parma), whom we sincerely thank.
**Acknowledgments**
The research leading to this paper has received funding through the CLARA project from the EU's Horizon
2020 research and innovation programme under the Grant Agreement No 730482.
Authors acknowledge with gratitude Daniela Molinari, who provided the data for the Adda case study,
within the framework of the Flood-Impat+ project, funded by Fondazione Cariplo.

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
