# Peer review of "Testing empirical and synthetic flood damage models: the case of"

_Natural Hazards and Earth System Sciences, 2018_

## Referee Comment (RC1) · Anonymous Referee #1 · 12 Dec 2018

I have now read the article titled "Estimating flood damage in Italy: empirical vs expert-based modelling approach". The article focuses on the comparison of different models (empirical vs expert based and Multi-, Bi and Univariate models aiming at the estimation of flood losses in Italy. Given the plethora of models and approaches in the field the paper is important and interesting. Furthermore, the paper is well structured and written. I recommend it for publication following minor revision. Please consider the following comments before publication:

1. The title should be revised and become more attractive. How about: "Putting flood loss models to the test: the case of Italy" or something like that….(just a suggestion)

2. Chapter materials and methods: 3.1 data description – consider a few introductory sentences before listing the datasets used for the study.

3. Subchapter 3.2: This is a chapter full of dense information. I would prefer two chapters instead: one, giving an overview of the existing models and explaining their characteristics and, two, a chapter describing the method used by the authors focusing on the reasons why they chose to test the particular models.

4. In the proposed "method" chapter a schematic description of the model used or work flow would be good and very practical for the reader (a figure showing the models used, the category they belong to expert-based/empirical and UVM, BVM or MVM or a table with a short description of the models and their characteristics).

5. Page 8, line 4: "exposure indicators" why are these "exposure" and not "vulnerability" indicators?

6. Page 8, Table 1. What is "finishing level"?

7. Page 9, line 16: Age and heat system are not in table 1. If you do not use them do not mention them at all.

8. Is "number of floors" named "FN" as in table 1 or "NF" as in Figures 4 and 7?

9. The language is overall good. There are, however, some small typos that have to be edited. E.g. page 9, line 23: "such as high prediction accuracy" and not "such prediction accuracy".

10. Page 14, line 17: "micro-scale". What is considered a micro-, meso- and macro-scale? The issue of scale should be further discussed in the discussion chapter and conclusions.

11. Page 14, lines 18-19: the authors refer to one of the case study areas and suggest that the differences in the model results may be subject to the different type of flood that these areas experienced. This issue should be further discussed. Where all the events similar? What is the difference of the impact of a flash flood? What about the presence of debris? Are these models reliable for all these types of processes?

Please also note the supplement to this comment:
https://www.nat-hazards-earth-syst-sci-discuss.net/nhess-2018-324/nhess-2018-324-RC1-supplement.pdf

---

## Referee Comment (RC2) · Anonymous Referee #2 · 19 Dec 2018

The manuscript "Estimating flood damage in Italy: Empirical vs expert-based modelling approach" validates different types of flood damage models for Italy and discusses the advantages and disadvantages of these models. This is a very interesting paper and the most extensive comparison of flood damage models for a specific area I have seen so far. I therefore believe this paper is a useful contribution to the scientific literature. I do however have some comments/questions regarding the setup of the study and some discussion points to be considered.

More important points:

-Currently the data-driven models developed in this study have been produced with data points from the same event it is validated on, hence no model transfer of the data-driven models is included. In practice a model transfer from one event to another is

always required for flood risk studies, it would therefore be fairer to always train the models on 2 events and validate it on the third event. Such an approach is also carried out in Schröter et al., (2014) and Wagenaar et al., (2018) and both studies show that multi-variable models typically have more difficulties in such a transfer setting.

-I think the data-driven UVMs wouldn't perform so well in a transfer setting because the main advantage of MVMs seems their transferability (Wagenaar et al., 2018). In the current setup this advantage of MVMs isn't used. Also if the model setup is changed some discussion is required on how significant the model transfer is between the events and whether a MVM is required or whether the events are so similar that a UVM would do.

-For the wider applicability of the results of this research some more discussion is required on to what extend the good performing literature models are tailored to the specific flood event and setting. These expert-based models seem to be made for Italy and for similar flood events to the one seen in this study. Are these models for example also made for the same region, did the developers have access to the damage data of these events or did they carry out surveys in the region? Point here is to help the reader identify when you can take a model from the literature and when you can't and for this we need more information about the good performing literature models.

Minor points:

-The abstract currently mostly summarizes the method, as a reader I would be very curious about the findings (what works better). Could you summarize these in the abstract.

-Page 2 line 16-18: Can you clarify this sentence, it is unclear and seems very crucial for the story so I wouldn't want to look up the references to get this clarification.

- Page 3, line 32: You mention 1000 flood events in 45 years, that seems way too much, what do you mean here by the word "events"?

- Page 6, line 27: You choose to use relative flood damages rather than absolute flood damages. This is a common choice but I think not an obvious one, can you motivate this decision?

-Section 3.2 introduction: Nice overview on UVMs and MVMs but I think this needs something on the transferability advantage of MVMs (see above).

-Section 3.2.1: Can you make a heading for each literature model.

-Section 3.2.1: Huizinga got his damage curves from the literature also, could you reference to the study that Huizinga got his damage function from.

-Page 9, line 2: Change "observation" in "observed"

-The Random Forests and ANN both have all sorts of tuning parameters. Like number of neurons (ANN), minimum number of observations per leaf (RF), learning rate (ANN) and more. Could you describe how you determined these settings?

-On page 11, from line 20. You describe something about the setup of the study. I think this should be somewhere else in the manuscript as this probably applies to all data-driven models (that would be most fair to do this the same for all data-driven models). If not why did you do that differently for the other models?

-Sometimes you use the word "water velocity", sometimes "flow velocity" and sometimes "water flow velocity", I think commonly the word "flow velocity" is used. Can you unify this throughout the paper.

-Page 16, line 14. Not all these citations fit a root function to data they just all have damage curves that have the shape of a root function. So please rephrase the sentence before the citation (message can be the same).

-In this study a limited number of variables was available for the MVMs. If more variables had been available the models might have performed better. Can you make this point somewhere.

References:

Schröter, K., Kreibich, H., Vogel, K., Riggelsen, C., Scherbaum, F. and Merz, B.(2014), How useful are complex flood damage models? Water Resour. Res. 50, 3378–3395. doi:10.1002/2013WR014396, 2014.

Wagenaar, D., Lüdtke, S., Schröter, K., Bouwer, L., Kreibich, H., 2018. Regional and Temporal Transferability of Multivariable Flood Damage Models. Water Resources Research. Volume 54, Issue 1. https://doi.org/10.1029/2017WR022233

Please also note the supplement to this comment:
https://www.nat-hazards-earth-syst-sci-discuss.net/nhess-2018-324/nhess-2018-324-RC2-supplement.pdf

---

## Author Comment (AC1) · 28 Jan 2019

We would like to thank Reviewer 2 for the in-depth review of our paper. He helped us to imprive and clarify key points of the analysis and to make the discussion more valauble.

1. Currently the data-driven models developed in this study have been produced with data points from the same event it is validated on, hence no model transfer of the data-driven models is included. In practice a model transfer from one event to another is always required for flood risk studies, it would therefore be fairer to always train the models on 2 events and validate it on the third event. Such an approach is also carried out in Schröter et al., (2014) and Wagenaar et al., (2018) and both studies show that

multi-variable models typically have more difficulties in such a transfer setting.

Thank you for this very important comment. What the Reviewer suggests is definitely a valuable alternative for independent model validation. However, in case of adopting the suggested approach, one must consider that the results would depend on the selection of the calibrating events, since the available events are inevitably different in terms of data amount and quality. On the contrary, merging all the data and selecting two thirds in a Monte Carlo framework overtakes the problem of selecting one out of 3 available events. We believe this approach might increase the utility of the collected records and the statistical significance of the trained models. We added to 3.4 (page 9, Line 9): "Trained models share the same sampling approach for validation: the observation dataset is split in three parts, where two thirds are used to train the model and one third for validation. This process is iterated 1,000 times, scrabbling the data and resampling the training set at each cycle. The output takes the mean of all iterations and provides a curve which represents the empirical damage relationship for the three events. This cross-validation approach has been previously employed in Hasanzadeh Nafari et al. (2017) and in Seifert et al. (2010) in order to optimise the statistic utility of the collected sample. "

2. I think the data-driven UVMs wouldn't perform so well in a transfer setting because the main advantage of MVMs seems their transferability (Wagenaar et al., 2018). In the current setup this advantage of MVMs isn't used. Also if the model setup is changed some discussion is required on how significant the model transfer is between the events and whether a MVM is required or whether the events are so similar that a UVM would do.

As specified in 3.4 (now improved), all trained models share the same scrambling-and-resampling iterative approach. Changing the training approach for the UVM would mean to change it also for the MVMs in order for the comparison to remain mean-ingful. The advantage of MVMs is that they consider location-specific indicators and more hazard variables in addition to water depth; by feeding the MVMs with these

event-specific data (10 variables), while UVM only consider water depth, we are in fact assessing the added value of MVM and thus their transferability potential. See also the previous comment on that.

3. For the wider applicability of the results of this research some more discussion is required on to what extend the good performing literature models are tailored to the specific flood event and setting. These expert-based models seem to be made for Italy and for similar flood events to the one seen in this study. Are these models for example also made for the same region, did the developers have access to the damage data of these events or did they carry out surveys in the region? Point here is to help the reader identify when you can take a model from the literature and when you can't and for this we need more information about the good performing literature models.

Thank you for pointing out this. More details have been added to the description of literature models and the source of their data. Also, additional explanations have been added in the discussion section.

4. The abstract currently mostly summarizes the method, as a reader I would be very curious about the findings (what works better). Could you summarize these in the abstract.

Thank you, we updated the abstracts with more details about the findings.

5. Page 2 line 16-18: Can you clarify this sentence, it is unclear and seems very crucial for the story so I wouldn't want to look up the references to get this clarification.

The sentence has been rewritten as: "Synthetic models, on the other hand, are based on "what-if analyses", relying on expert-based knowledge in order to generalise the relation between the magnitude of a hazard event and the resulting damage estimate. That means, synthetic models have a higher level of standardisation and thus are better suited for both temporal and spatial transferability."

6. Page 3, line 32: You mention 1000 flood events in 45 years, that seems way too

much, what do you mean here by the word "events"?

Correct observation, the number of events refer to the AVI catalogue from CNR and in their records there are more than 1,000 unique event codes, however some of them refer to the same date. We then aggregated events in the same date and corrected the number to 300 events.

7. Page 6, line 27: You choose to use relative flood damages rather than absolute flood damages. This is a common choice, but I think not an obvious one, can you motivate this decision?

Added: "We chose to measure impacts in relative terms so as to make them easier to compare through different times (inflation effect) and places (different currencies)."

8. Section 3.2 introduction: Nice overview on UVMs and MVMs but I think this needs something on the transferability advantage of MVMs (see above).

Improved the intro: "[. . .] other parameters may influence the flood damage process, [...] a large number of other non-hazard factors can be significantly different from one place to another [...] Multivariable models (MVMs) can account for such additional factors and thus are able to adapt the damage estimate to the characteristics of a specific event and location. Therefore, they may be better-suited to describe the complexity of the flood-damage process for transferability purpose."

9. Section 3.2.1: Can you make a heading for each literature model.

Sub-chapters have been split differently to improve readibility.

10. Section 3.2.1: Huizinga got his damage curves from the literature also, could you reference to the study that Huizinga got his damage function from.

That is quite a long list of studies that have been averaged, none of which related to Italy; for this reason, we prefer to keep it shorter.

11. Page 9, line 2: Change "observation" in "observed"

Changed "observation datasets" into "observed records".

12. The Random Forests and ANN both have all sorts of tuning parameters. Like number of neurons (ANN), minimum number of observations per leaf (RF), learning rate (ANN) and more. Could you describe how you determined these settings?

Unless specified, RF and ANN run on default parameters. We added the minimum number of observations per leaf in RF (5). We also added to ANN: "The learning rate is controlled by coefficient $\mu$: when $\mu$ is very small, the training process approximates the Gauss-Newton optimization algorithm (i.e. fast learning, low stability), while when $\mu$ is very large, the training process resembles the steepest descent algorithm (i.e. slow learning, high stability). The value of $\mu$ starts as 1 and is updated during each training epoch. In case a training epoch is successful in reducing the SSE in the output layer, then $\mu$ is reduced by half; otherwise, the value of $\mu$ is increased by a factor of two and a new training attempt is performed." The number of neurons in ANN is already specified: "the initial number of hidden neurons per hidden layer is approximated as two-thirds of the summation of the number of neurons in the previous and next layers".

13. On page 11, from line 20. You describe something about the setup of the study. I think this should be somewhere else in the manuscript as this probably applies to all data-driven models (that would be most fair to do this the same for all data-driven models). If not why did you do that differently for the other models?

The referenced setup is specifically related to the ANN model; we explained better the training procedure that is shared among the trained models (3.4, pg 9 line 11:) "All these models share the same sampling approach for training and validation: the observation dataset is split in three parts, where two thirds are used to train the model and one third for validation. This process is iterated 1,000 times, scrabbling the data and resampling the training set at each cycle."

14. Sometimes you use the word "water velocity", sometimes "flow velocity" and sometimes "water flow velocity", I think commonly the word "flow velocity" is used. Can you

unify this throughout the paper.

Yes, thank you, we now use "Flow velocity" consistently.

15. Page 16, line 14. Not all these citations fit a root function to data they just all have damage curves that have the shape of a root function. So please rephrase the sentence before the citation (message can be the same).

Thanks for this comment. The sentence has been rephrased as the following: "Our findings confirm previous results indicating that the curve shape described by the root function is the most adequate to describe the flood damage process".

16. In this study a limited number of variables was available for the MVMs. If more variables had been available the models might have performed better. Can you make this point somewhere.

Added to discussion: "We can't exclude that the performances of MVMs would benefit from the inclusion of additional predictive variables, such as those related to the early warning system and precaution measures, or social vulnerability; however, the availability of such information is limited for our case study."

17. References: Schröter, K., Kreibich, H., Vogel, K., Riggelsen, C., Scherbaum, F. and Merz, B. (2014), How useful are complex flood damage models? Water Resour. Res. 50, 3378–3395. doi:10.1002/2013WR014396, 2014. Wagenaar, D., Lüdtke, S., Schröter, K., Bouwer, L., Kreibich, H., 2018. Regional and Temporal Transferability of Multivariable Flood Damage Models. Water Resources Research. Volume 54, Issue 1. https://doi.org/10.1029/2017WR022233

Interesting articles, thank you, these have been added to the discussion.

Please also note the supplement to this comment:
https://www.nat-hazards-earth-syst-sci-discuss.net/nhess-2018-324/nhess-2018-324-AC1-supplement.pdf

---

## Author Comment (AC2) · 28 Jan 2019

We would like to thank Reviewer 1 for the helpful comments and suggestions that made our paper more consistent and readable.

1. The title should be revised and become more attractive. How about: "Putting flood loss models to the test: the case of Italy" or something like that....(just a suggestion)

Thank you, the title has been revised as "Testing empirical and synthetic flood damage models: the case of Italy"

2. Chapter materials and methods: 3.1 data description – consider a few introductory sentences before listing the datasets used for the study.

[Figure]

Added: "Our purpose is first to draw a detailed, homogeneous description of the hazard and exposure features involved in the three hazard events in order to evaluate their relationship with measured impacts. Several datasets are required for this task. These have been collected from different sources and spatially projected to the building level (i.e. micro-scale) for each one of the three study areas. The dataset we compiled for this analysis comprises:"

3. Subchapter 3.2: This is a chapter full of dense information. I would prefer two chapters instead: one, giving an overview of the existing models and explaining their characteristics and, two, a chapter describing the method used by the authors focusing on the reasons why they chose to test the particular models.

To improve readability 3.2 has been split into 3 sub-chapters (3.2. Damage models overview; 3.3 Models from Literature; 3.4 Models trained on observed records)

4. In the proposed "method" chapter a schematic description of the model used or work flow would be good and very practical for the reader (a figure showing the models used, the category they belong to expert-based/empirical and UVM, BVM or MVM or a table with a short description of the models and their characteristics).

A workflow figure has been added as 3.4.3.

5. Page 8, line 4: "exposure indicators" why are these "exposure" and not "vulnerability" indicators?

Clarified: "Indicators related to exposure and vulnerability"

6. Page 8, Table 1. What is "finishing level"?

Finishing level represent the state of quality of a buildings, as described in INSYDE details.

7. Page 9, line 16: Age and heat system are not in table 1. If you do not use them do not mention them at all.

Correct, they were deleted.

8. Is "number of floors" named "FN" as in table 1 or "NF" as in Figures 4 and 7?

NF is the right acronym. Thanks for having spotted it, the revised version is now consistent.

9. The language is overall good. There are, however, some small typos that have to be edited. E.g. page 9, line 23: "such as high prediction accuracy" and not "such prediction accuracy".

Thank you. We checked the overall manuscript with the help of a professional translator, we hope to have fixed all the typos.

10. Page 14, line 17: "micro-scale". What is considered a micro-, meso- and macro-scale? The issue of scale should be further discussed in the discussion chapter and conclusions.

Added in page 7, line 18: "Models can further be classified in relation to the scale of their development and application (de Moel et al., 2015): "micro-scale" usually refers to a model built to account impacts over buildings individual components and it is commonly applied for local assessment; "meso-scale" refers to sub-national analysis which commonly relies on data aggregated on provincial or regional administrative units; "macro-scale" concerns assessments at country level." Added specification of scale in conclusions.

11. Page 14, lines 18-19: the authors refer to one of the case study areas and suggest that the differences in the model results may be subject to the different type of flood that these areas experienced. This issue should be further discussed. Where all the events similar? What is the difference of the impact of a flash flood? What about the presence of debris? Are these models reliable for all these types of processes?

Added explanation: "In fact, Luino's model was produced based on a flash flood event characterised by higher flow velocities and larges relative impacts". In all other cases,

we speak of river floods and not flash floods, we specified in text. Also added in the conclusion: "The results have shown important errors when transferring models derived from different countries and scales such as the JRC-IT curve, or from events with different characteristics: the model from Luino is based on a flash-flood event where flow velocity has likely a significant role on the event impact."

Please also note the supplement to this comment:
https://www.nat-hazards-earth-syst-sci-discuss.net/nhess-2018-324/nhess-2018-324-AC2-supplement.pdf

**Supplement:**

**Reviewer #1**

I have now read the article titled "Estimating flood damage in Italy: empirical vs expert-based modelling approach". The article focuses on the comparison of different models (empirical vs expert based and Multi-, Bi and Univariate models aiming at the estimation of flood losses in Italy. Given the plethora of models and approaches in the field the paper is important and interesting. Furthermore, the paper is well structured and written. I recommend it for publication following minor revision. Please consider the following comments before publication:

1. The title should be revised and become more attractive. How about: "Putting flood loss models to the test: the case of Italy" or something like that….(just a suggestion)

Thank you, the title has been revised as "Testing empirical and synthetic flood damage models: the case of Italy"

2. Chapter materials and methods: 3.1 data description – consider a few introductory sentences before listing the datasets used for the study.

> Added: "Our purpose is first to draw a detailed, homogeneous description of the hazard and exposure features involved in the three hazard events in order to evaluate their relationship with measured impacts. Several datasets are required for this task. These have been collected from different sources and spatially projected to the building level (i.e. micro-scale) for each one of the three study areas. The dataset we compiled for this analysis comprises:"

3. Subchapter 3.2: This is a chapter full of dense information. I would prefer two chapters instead: one, giving an overview of the existing models and explaining their characteristics and, two, a chapter describing the method used by the authors focusing on the reasons why they chose to test the particular models.

> 3.2 has been split into 3 sub-chapters (3.2. Damage models overview; 3.3 Models from Literature; 3.4 Models trained on observed records)

4. In the proposed "method" chapter a schematic description of the model used or work flow would be good and very practical for the reader (a figure showing the models used, the category they belong to expert-based/empirical and UVM, BVM or MVM or a table with a short description of the models and their characteristics).

> A workflow figure has been added as 3.4.3.

5. Page 8, line 4: "exposure indicators" why are these "exposure" and not "vulnerability" indicators?

> "Indicators related to exposure and vulnerability"

6. Page 8, Table 1. What is "finishing level"?

> Finishing level represent the state of quality of a buildings, as described in INSYDE.

7. Page 9, line 16: Age and heat system are not in table 1. If you do not use them do not mention them at all.

> ➢ deleted

8. Is "number of floors" named "FN" as in table 1 or "NF" as in Figures 4 and 7?

> ➢ NF is the right acronym. Thanks for having spotted it, the revised version is now consistent.

9. The language is overall good. There are, however, some small typos that have to be edited. E.g. page 9, line 23: "such as high prediction accuracy" and not "such prediction accuracy".

> ➢ Thank you. We checked the overall manuscript and we hope to have fixed all the typos.

10. Page 14, line 17: "micro-scale". What is considered a micro-, meso- and macro-scale? The issue of scale should be further discussed in the discussion chapter and conclusions.

> ➢ Added in page 7, line 18: "Models can further be classified in relation to the scale of their development and application (de Moel et al., 2015): "micro-scale" usually refers to a model built to account impacts over buildings individual components and it is commonly applied for local assessment; "meso-scale" refers to sub-national analysis which commonly relies on data aggregated on provincial or regional administrative units; "macro-scale" concerns assessments at country level." Added specification of scale in conclusions.

11. Page 14, lines 18-19: the authors refer to one of the case study areas and suggest that the differences in the model results may be subject to the different type of flood that these areas experienced. This issue should be further discussed. Where all the events similar? What is the difference of the impact of a flash flood? What about the presence of debris? Are these models reliable for all these types of processes?

> ➢ Added explanation: "In fact, Luino's model was produced based on a flash flood event characterised by higher flow velocities and larges relative impacts". In all other cases, we speak of river floods and not flash floods, we specified in text. Also added in the conclusion: "The results have shown important errors when transferring models derived from different countries and scales such as the JRC-IT curve, or from events with different characteristics: the model from Luino is based on a flash-flood event where flow velocity has likely a significant role on the event impact."

**Reviewer #2**

The manuscript "Estimating flood damage in Italy: Empirical vs expert-based modelling approach" validates different types of flood damage models for Italy and discusses the advantages and disadvantages of these models. This is a very interesting paper and the most extensive comparison of flood damage models for a specific area I have seen so far. I therefore believe this paper is a useful contribution to the scientific literature. I do however have some comments/questions regarding the setup of the study and some discussion points to be considered.

**More important points:**

• Currently the data-driven models developed in this study have been produced with data points from the same event it is validated on, hence no model transfer of the data-driven models is included. In practice a model transfer from one event to another is always

required for flood risk studies, it would therefore be fairer to always train the models on 2 events and validate it on the third event. Such an approach is also carried out in Schröter et al., (2014) and Wagenaar et al., (2018) and both studies show that multi-variable models typically have more difficulties in such a transfer setting.

➢ Thank you for this very important comment. What the Reviewer suggests is definitely a valuable alternative for independent model validation. However, in case of adopting the suggested approach, one must consider that the results would depend on the selection of the calibrating events, since the available events are inevitably different in terms of data amount and quality. On the contrary, merging all the data and selecting two thirds in a Monte Carlo framework overtakes the problem of selecting one out of 3 available events. We believe this approach might increase the utility of the collected records and the statistical significance of the trained models.

➢ Added to 3.4 (page 9, Line 9):

Trained models share the same sampling approach for validation: the observation dataset is split in three parts, where two thirds are used to train the model and one third for validation. This process is iterated 1,000 times, scrabbling the data and resampling the training set at each cycle. The output takes the mean of all iterations and provides a curve which represents the empirical damage relationship for the three events. This cross-validation approach has been previously employed in Hasanzadeh Nafari et al. (2017) and in Seifert et al. (2010) in order to optimise the statistic utility of the collected sample.

• I think the data-driven UVMs wouldn't perform so well in a transfer setting because the main advantage of MVMs seems their transferability (Wagenaar et al., 2018). In the current setup this advantage of MVMs isn't used. Also if the model setup is changed some discussion is required on how significant the model transfer is between the events and whether a MVM is required or whether the events are so similar that a UVM would do.

➢ As specified in 3.4 (now improved), all trained models share the same scrambling-and-resampling iterative approach. Changing the training approach for the UVM would mean to change it also for the MVMs in order for the comparison to remain meaningful. The advantage of MVMs is that they consider location-specific indicators and more hazard variables in addition to water depth; by feeding the MVMs with these event-specific data (10 variables), while UVM only consider water depth, we are exactly assessing the added value of MVM and thus their transferability potential. See also the previous comment on that.

• For the wider applicability of the results of this research some more discussion is required on to what extend the good performing literature models are tailored to the specific flood event and setting. These expert-based models seem to be made for Italy and for similar flood events to the one seen in this study. Are these models for example also made for the same region, did the developers have access to the damage data of these events or did they carry out surveys in the region? Point here is to help the reader identify when you can take a model from the literature and when you can't and for this we need more information about the good performing literature models.

➢ Thank you for pointing out this. More details have been added to the description of literature models and the source of their data. Also, additional explanations have been added in the discussion section.

**Minor points:**

• The abstract currently mostly summarizes the method, as a reader I would be very curious about the findings (what works better). Could you summarize these in the abstract.

➢ Thank you, we updated the abstracts with details about the findings.

• Page 2 line 16-18: Can you clarify this sentence, it is unclear and seems very crucial for the story so I wouldn't want to look up the references to get this clarification.

➢ The sentence has been rewritten as: "Synthetic models, on the other hand, are based on ''what-if analyses'', relying on expert-based knowledge in order to generalise the relation between the magnitude of a hazard event and the resulting damage estimate. That means, synthetic models have a higher level of standardisation and thus are better suited for both temporal and spatial transferability."

• Page 3, line 32: You mention 1000 flood events in 45 years, that seems way too much, what do you mean here by the word "events"?

➢ Correct observation, the number of events refer to the AVI catalogue from CNR and in their records there are more than 1,000 unique event codes, however some of them refer to the same date. We then aggregated events in the same date and corrected the number to 300 events.

• Page 6, line 27: You choose to use relative flood damages rather than absolute flood damages. This is a common choice, but I think not an obvious one, can you motivate this decision?

➢ We chose to measure impacts in relative terms so to make them easier to compare through different times (inflation effect) and places (different currencies).

• Section 3.2 introduction: Nice overview on UVMs and MVMs but I think this needs something on the transferability advantage of MVMs (see above).

➢ Improved the intro:
"[…] other parameters may influence the flood damage process, [...] a large number of other non-hazard factors can be significantly different from one place to another [...] Multivariable models (MVMs) can account for such additional factors and thus are able to adapt the damage estimate to the characteristics of a specific event and location. Therefore, they may be better-suited to describe the complexity of the flood-damage process for transferability purpose."

• Section 3.2.1: Can you make a heading for each literature model.

➢ Sub-chapters have been split differently

• Section 3.2.1: Huizinga got his damage curves from the literature also, could you reference to the study that Huizinga got his damage function from.

  ➢ That's quite a long list of studies that have been averaged, none of which related to Italy; for this reason, we prefer to keep it shorter.

• Page 9, line 2: Change "observation" in "observed"

  ➢ Changed "observation datasets" into "observed records"

• The Random Forests and ANN both have all sorts of tuning parameters. Like number of neurons (ANN), minimum number of observations per leaf (RF), learning rate (ANN) and more. Could you describe how you determined these settings?

  ➢ Unless specified, RF and ANN run on default parameters. We added the minimum number of observations per leaf in RF (5). We also added to ANN: "The learning rate is controlled by coefficient $\mu$: when $\mu$ is very small, the training process approximates the Gauss-Newton optimization algorithm (i.e. fast learning, low stability), while when $\mu$ is very large, the training process resembles the steepest descent algorithm (i.e. slow learning, high stability) (Wilamowski & Irwin, 2011). The value of $\mu$ starts as 1 and is updated during each training epoch. In case a training epoch is successful in reducing the SSE in the output layer, then $\mu$ is reduced by half; otherwise, the value of $\mu$ is increased by a factor of two and a new training attempt is performed."
    The number of neurons in ANN is already specified: "the initial number of hidden neurons per hidden layer is approximated as two-thirds of the summation of the number of neurons in the previous and next layers".

• On page 11, from line 20. You describe something about the setup of the study. I think this should be somewhere else in the manuscript as this probably applies to all data-driven models (that would be most fair to do this the same for all data-driven models). If not why did you do that differently for the other models?

  ➢ The referenced setup is specifically related to the ANN model; we explained better the training procedure that is shared among the trained models (3.4, pg 9 line 11:) "All these models share the same sampling approach for training and validation: the observation dataset is split in three parts, where two thirds are used to train the model and one third for validation. This process is iterated 1,000 times, scrabbling the data and resampling the training set at each cycle."

• Sometimes you use the word "water velocity", sometimes "flow velocity" and sometimes "water flow velocity", I think commonly the word "flow velocity" is used. Can you unify this throughout the paper.

  ➢ Yes, thank you

• Page 16, line 14. Not all these citations fit a root function to data they just all have damage curves that have the shape of a root function. So please rephrase the sentence before the citation (message can be the same).

> Thanks for the comment. The sentence has been rephrased as the following: "Our findings confirm previous results indicating that the curve shape described by the root function is the most adequate to describe the flood damage process".

• In this study a limited number of variables was available for the MVMs. If more variables had been available the models might have performed better. Can you make this point somewhere.

> Added to discussion: "We can't exclude that the performances of MVMs would benefit from the inclusion of additional predictive variables, such as those related to the early warning system and precaution measures, or social vulnerability; however, the availability of such information is limited for our case study."

Schröter, K., Kreibich, H., Vogel, K., Riggelsen, C., Scherbaum, F. and Merz, B. (2014), How useful are complex flood damage models? Water Resour. Res. 50, 3378–3395. doi:10.1002/2013WR014396, 2014.

Wagenaar, D., Lüdtke, S., Schröter, K., Bouwer, L., Kreibich, H., 2018. Regional and Temporal Transferability of Multivariable Flood Damage Models. Water Resources Research. Volume 54, Issue 1. https://doi.org/10.1029/2017WR022233

> Very interesting thank you, these have been added to the discussion.